# Investigating the Modulation of the VTA Neurons in Nicotine-Exposed Pups during Early Maturation Using Optogenetics

**DOI:** 10.3390/ijms24032280

**Published:** 2023-01-23

**Authors:** Austin Ganaway, Yoshinori Sunaga, Yasumi Ohta, Jun Ohta, Metin Akay, Yasemin M. Akay

**Affiliations:** 1Biomedical Engineering Department, University of Houston, 3517 Cullen Blvd, Houston, TX 77204, USA; 2Division of Materials Science, Graduate School of Science and Technology, Nara Institute of Science and Technology, 8916-5 Takayama, Ikoma 630-0192, Nara, Japan

**Keywords:** nicotine, dopamine, VTA, optogenetics, ChrimsonR, microdialysis, NAc, photo-stimulation, maturation

## Abstract

Advancing the understanding of the relationship between perinatal nicotine addiction and the reward mechanism of the brain is crucial for uncovering and implementing new treatments for addiction control and prevention. The mesolimbic pathway of the brain, also known as the reward pathway, consists of two main areas that regulate dopamine (DA) and addiction-related behaviors. The ventral tegmental area (VTA) releases DA when stimulated, causing the propagation of neuronal firing along the pathway. This ends in the release of DA into the extracellular space of the nucleus accumbens (NAc), which is directly modulated by the uptake of DA. Much research has been conducted on the effects of nicotine addiction, but little research has been conducted concerning nicotine addiction and the mesolimbic pathway regarding maturation due to the small brain size. In this study, we apply our novel microstimulation experimental system to rat pups that have been perinatally exposed to nicotine. By using our self-fabricated photo-stimulation (PS) device, we can stimulate the VTA and collect dialysate, which is then used to estimate DA released into the NAc. The proposed platform has demonstrated the potential to monitor neural pathways as the pups mature.

## 1. Introduction

The mesolimbic system, known as the reward circuit, is a complex neural network used for the release and transport of dopamine (DA), leading primarily from the ventral tegmental area (VTA) to the nucleus accumbens (NAc). Dopaminergic neurons are found largely within the VTA, and they are responsible for the proliferation of DA, which is the main proponent in the regulation of reward behavior. Addictive substances, such as nicotine and alcohol, directly modulate the regulation of DA and result in both an increase in DA release and increased uptake of DA by the NAc. A feedback system is reinforced upon uptake, leading to the potential formation of a detrimental cycle of substance abuse [1]. Substance abuse remains a critical issue in modern society, with lasting effects for both users and those exposed in a second-hand manner. Considering the widespread use of cigarettes and, more recently, electronic cigarettes, the risk of perinatal nicotine addiction is an ongoing issue, with 1 out of 14 women reporting smoking during pregnancy in the United States alone [2]. The outcome of perinatal nicotine exposure via smoking is possibly severe, with detrimental health defects for the fetus, including low birth weight, preterm delivery, and stillbirth, which may result in many complications, including developmental delays [3]. Additionally, the dysregulation of the release of DA is linked to many neurological diseases, including attention deficit hyperactivity disorder (ADHD), schizophrenia, and psychosis [4,5]. It is crucial to understand how perinatal nicotine exposure affects the maturation of the mesolimbic pathway to assess better and treat both nicotine-exposed mothers and children.

To better understand the effects of the mesolimbic pathway on DA regulation with regards to maturation, it is crucial to understand the activation patterns of the DA neurons. The DA neurons within the VTA are known to exhibit spontaneous firing action which reflects a sequence of alternating patterns [6]. These patterns include a regular spike firing and burst firing. Although regular spike firing is thought to be used to suppress neuronal signaling related to inconsequential stimulation, burst firing of the neurons is hypothesized to play a critical role in extracellular DA release. Intense release of DA is most likely caused by the higher firing frequencies [6].

In our previous study, we characterized the effects of prenatal nicotine exposure on the mesocorticolimbic system of the rat offspring, where local field potentials were recorded from 27 sites across the VTA of 9 rats aged 40–55 days [7]. The extracellular VTA neural activities were analyzed using the Approximate Entropy (ApEn) method. Approximate entropy values were then grouped according to each anatomic location including the subsections of the VTA, the parabrachial pigmented nucleus (PBP), the parainterfascicular nucleus (PIF), and the paranigral nucleus (PN). Our results show that the local field potentials corresponding to the neurons located in the PIF region of the VTA have ApEn values significantly higher in the maternal nicotine cases when compared to the control. Therefore, we concluded that the dopamine neurons located in the PIF sub-region of the VTA are very likely involved with the nicotine addiction.

It is understood that neurons in the VTA activate when chronically exposed to nicotine, leading to propagated neuronal burst firing along the projection pathways of the mesocorticolimbic system [7,8]. The VTA is composed of DA, γ-aminobutyric acid (GABA), and glutamate neurons, and, as shown in previous studies, using the binding of nicotine and the abundant nicotinic acetylcholine receptors (nAChRs) within the VTA, nicotine can activate DA neurons directly while activating GABA and glutamate neurons indirectly [9,10]. This induces a robust release of DA to the NAc.

The study of dopaminergic and GABAergic pathways has led to a better understanding of neurochemical changes and the resulting effects when introducing stimuli to adult rats; however, little research has been conducted concerning the effects of addiction regarding maturation and the release and distribution of neurotransmitters, specifically with perinatal nicotine addiction [11]. Thus, monitoring the mesolimbic system during maturation is crucial to gleaning new information, and observation of rats during the early days of development is necessary. The challenge resides in the size of the juvenile rat brain, making it problematic to access difficult-to-reach locations, such as the VTA, located deep within the midbrain. To overcome these difficulties, we propose to use a micro-stimulation device with microdialysis to investigate the neural networks of maturing rats.

In a previous study, we implemented complementary metal-oxide semiconductor (CMOS) imaging of the VTA in conjunction with microdialysis of the NAc and prefrontal cortex (PFC), evaluated using high-performance liquid chromatography with electrochemical detection (HPLC-ECD), using GCaMP transgenic mice to monitor DA neural activities. By combining GCaMP fluorescence of the VTA and dialysate from the terminal points of the neural pathways, the DA release related to nicotine intake was simultaneously measured. Our results indicate that our novel platform is capable of circumventing traditional stimulation and imaging methods by taking advantage of microimaging and microstimulation implantable devices to simultaneously monitor DA release and uptake [12].

In a more recent study, we developed and implemented a novel stimulation system designed for adult rats and monkeys with the intention of stimulating large areas of the brain using a photo-stimulation (PS) device. Additionally, a microdialysis probe was implanted to provide real-time analysis of the DA concentration within the PFC and NAc. The data obtained suggest this novel PS device is capable of stimulating large areas of the brain with an irradiation area of 1–2 mm of depth. Additionally, using this system we were able to detect the increased response of dopaminergic neurons within the deep brain, allowing for an estimation of DA concentration. Our platform demonstrates the innate advantage of our novel PS device when compared to traditional optical fiber technology typically used for optogenetics experimentation due to its ease of implementation and flexibility. Expression of the adeno-associated virus (AAV) was confirmed following the success of stimulation and dialysate collection and analysis, demonstrating the efficacy of the novel system. Channelrhodopsin-2 (ChR2) was expressed in the VTA of macaque monkeys and ChrimsonR was expressed in the VTA of rats. It was noted that, due to increased absorption of light by brain tissue at lower wavelengths, ChR2 posed more risk for cells relating to phototoxicity and heat damage than ChrimsonR due to the requirement of blue excitation light [13]. Both animal types demonstrated successful stimulation and estimation of DA, and the ChrimsonR AAV, which takes advantage of red excitation light that has a higher wavelength and lower absorption coefficient, expressed in the rat VTA showed encouraging results concerning DA projection [14].

Using our novel platform, our most recent study investigated the relationship between GABA neurons and dopaminergic neurons. To do so, the DA release in the NAc of adult mice was quantified following stimulation by implementing the previously used PS device into the VTA in tandem with a microdialysis system. This system was chosen due to its unconstrained properties compared to more traditional stimulation methods, allowing for freely moving experimental conditions that better emulate a natural setting. Both wild-type mice with ChrimsonR AAV expressed in the VTA and dopamine transporter (DAT)-Cre (dopamine specific) transgenic mice with ChrimsonR AAV expressed in the VTA were used. Our findings demonstrate that the novel PS device is capable of effectively and consistently stimulating the VTA and the microdialysis system can provide accurate estimations of DA concentration within the NAc. It was concluded that it is capable of detecting the increase and decrease in DA release following the excitation of dopaminergic neurons and GABAergic neurons [15].

In this study, we investigate the influence of perinatal substance abuse on newborn pups by stimulating the VTA neurons of juvenile rats with our novel optogenetics system and by recording DA release in the NAc using microdialysis. We chose to implement optogenetics stimulation given that optogenetics techniques have been recently used as a catalyst for mapping and understanding the dynamics of neural activities. We evaluate the efficacy of this experimental system by applying it to nicotine-exposed pups from P14 onward. We then analyzed the trends of DA release in the NAc obtained through microdialysis following stimulation of the VTA.

## 2. Results

The initial results obtained from the three representative nicotine-exposed pups are presented in (Figure 1). The time-lapse results of the HPLC-ECD analyzed samples relay an increase or decrease in DA concentration 15 min after stimulation. The P14 nicotine-exposed pup in (Figure 1A) demonstrated a noticeable increase in DA release into the NAc, which signifies that ChrimsonR was activated via the PS device, leading to the excitation of both GABAergic and dopaminergic neurons. Following the initial slight in DA release increase at 15 min in the NAc, the increase is more pronounced at the 30 min sample. Then, the DA concentration begins to return to the control level. For the P21 nicotine-exposed pup, shown in (Figure 1B), there is a gradual increase in the DA release over time following the initiation of the PS device, with an increase occurring after 30 min. In (Figure 1C), the P28 pup showed an increase at the 30 min sample, followed by a gradual decline to the control value.

After successfully stimulating and obtaining DA data from the representative P14, P21, and P28 pups, we stimulated and obtained the DA release for multiple animals. Figure 2 summarizes the results of DA releases in the NAc following the stimulations in six nicotine-exposed pups. Please note that for each pup, we stimulated at least twice. The results showed a significant difference in the quantity of DA released, specifically between the P14 animals and the P21 animals (*p*-values: control—7.09 × 10^−7^, 15 min—0.0106, 30 min—0.0019, 45 min—0.0135, 60 min—0.00689). While rats at age P21 and P28 have similar output levels, the younger pups, at P14, produced a significantly lower amount of DA after stimulation. Please also note that the difference between P21 and P28 was not statistically significant.

While undergoing experimentation on nicotine-exposed pups, a noticeable drop in DA immediately following optogenetic stimulation was observed in two P14 animals. As seen in (Figure 3), a nicotine-exposed P14 pup and a control P14 pup both exhibited a drop in extracellular DA concentration in the NAc at the 15-min sample. After the drop, a rise in DA concentration is observed at the 30-min mark. Following this, the animal proceeds to demonstrate the expected characteristic DA output trend. It can then be hypothesized that this drop occurs due to the stimulation of both dopaminergic and GABAergic neurons in the VTA.

Finally, to investigate the influence of nicotine exposure on DA release, we used two new nicotine-exposed and control P14 pups. Please note that both animals are stimulated at least twice. The results indicate that stimulus significantly increased DA release in nicotine-exposed animals compared to control, specifically at the 15-min and 30-min sample (*p*-value of 0.049 and 0.045, respectively). Therefore, it can be extrapolated that the maturation of the nicotine-exposed pups has been fundamentally altered to that of the control pups.

## 3. Discussion

In this study, we modified our stimulation platform designed for adult mice to stimulate the VTA and collect and analyze DA release in the NAc of rat pups [15]. The same system was used for adult mice, and we adapted it to rat pups, starting at P14. For the P14 pup from (Figure 1A), the DA concentration following stimulation using the PS device produced a considerable reaction, noted by the increase in DA released to the NAc at the 30 min sample, with an increase (20%) in reference to the control. As shown by the P21 pup in (Figure 1B), we observed a consistent response to stimulation due to the increase in DA release, followed by a steady decline back to the control value after stimulation. When observing the P28 pup, shown in (Figure 1C), there is an initial dip in DA concentration following stimulation. A noticeable increase (18%) occurs at the 30 min mark, followed by a steady decline to the control value.

Our preliminary studies showed that we successfully stimulated the VTA with an effective ChrimsonR AAV injection in conjunction with the implanted PS device based on data acquired in our previous work [14,15]. Furthermore, our study confirms the findings of previous studies based on ethanol and sucrose-exposed rats in which DA was obtained from the NAc [16], since their study also shows similar DA release patterns. Additionally, another research group also verified the feasibility of microdialysis and DA release by injecting nicotine into adult rats during freely moving experiments and obtaining samples from the extracellular space of the NAc [17]. After doing so, an increase in DA concentration occurred at the 20 min mark, followed by a steady decline to the baseline value. As for maturation, this trend is confirmed to be observable in both maturing and mature rats [18]. These studies encouraged us to investigate nicotine exposure on the DA release during early maturations.

We also observed that the average raw DA output of both P21 and P28 nicotine-exposed rats is significantly higher than that of the nicotine-exposed P14 pups. The increase in DA concentration occurs at the 30 min sample for the more mature animals, with the P21 pups showing the most robust release of DA. The DA output of P28 pups is more than twice that of P14 pups, and the release for the P21 pups proves to be almost three times those of the youngest pups. There are many potential causes for such a stark contrast in DA release. Most notably, the neural maturation process for the pups occurs at a different pace when compared to humans. For humans, the majority of synaptogenesis occurs before birth during the final trimester but, for rats, this process occurs during the first two weeks, post-gestation [19]. In other words, the final trimester of human gestation is equivalent to the days P0-P14 for rat pups [20]. Given the significant growth spurt of the brain concerning neuron proliferation during this period, it can be concluded that the post-gestational mechanism for increased axonal growth and gliogenesis is the main driving component in the increased release of DA in the NAc [21]. Additionally, it is important to note that the expression of the ChrimsonR AAV can play a role in the increased release of DA. Based on results published previously and the slightly decreased release of DA from the P28 pups in (Figure 2), we can hypothesize that the effect of further expression of the ChrimsonR AAV is negligible [14,15].

Gene expression of DA-related proteins is also a critical component to the establishment of healthy neural pathways and the resulting release of neurotransmitters via said pathways. The dopamine transporter (DAT) protein, otherwise known as the solute carrier family 6 member 3 (Slc6a3), is a commonly used DA marker responsible for the regulation of DA. It controls DA levels by regulating the re-uptake of DA found in extracellular space. DAT does so by gathering DA back into the presynaptic neuron, which is then sent out from the synapse into the cytosol. Previous studies have confirmed a significant increase in the expression of DAT as the animal matures from gestation day 15 (G15) to adulthood with a stark increase occurring between P7 and P17, making a more robust release of DA at older ages plausible [22]. Another notable DA marker, Tyrosine Hydroxylase (TH), is also a key component in the production of DA. Its role involves the synthesis of DA and has been previously reported to have increased activity as the animal matures with a notable increase occurring around the P14 mark [23]. Therefore, by revealing this significant contrast in DA levels we believe there is potential for more in-depth analysis using this system.

Our preliminary study includes only wild-type rats. Therefore, we must consider that we are interacting with GABA and DA because both GABAergic and dopaminergic neurons are stimulated when ChrimsonR is activated. When we observed the DA concentration output from the animals referenced in (Figure 1), we saw the expected result, with almost every pup showing an immediate increase in DA concentration in the NAc following stimulation. Additional data obtained from P14 rats, shown in (Figure 3), introduced an unexpected drop immediately after stimulation. It can be hypothesized that the PS device activated GABAergic neurons closest to the implantation, which downregulated the production of DA relative to the stabilized control value. Thus, GABA’s effects were stronger than DA’s, resulting in a delayed DA increase. The orange bars shown in (Figure 3) derive from a P14 nicotine-exposed pup that underwent our adapted experimental setup. We see a drop of over half of the control value following stimulation, and then the increase in DA concentration occurs at the 30 min sample at roughly 1.5 times the control level. The same is demonstrated by the green bars in (Figure 3) where a control P14 pup reaches its DA increase at the 30 min mark rather than immediately after stimulation due to more than a thirty percent drop from the control sample. This furthermore confirms our results shown by the P28 pup in (Figure 1C), demonstrating that this abrupt drop can occur despite the pup’s age. In future experiments, we can confirm our hypothesis regarding GABA’s influence on DA output using genetically modified DAT-Cre rats in conjunction with the ChrimsonR AAV to isolate and activate DA neurons. In doing so, we will be able to quantify DA release in the NAc without the effects of GABAergic neurons.

To compare stimulated DA release in the NAc between rat pups exposed to nicotine and control pups, we conducted a separate study on a separate group of P14 nicotine-exposed and control pups. For future investigation, it is crucial to obtain a foundational understanding of the lasting effects brought about by chronic nicotine exposure during maturation. As seen in (Figure 4), the data obtained provide evidence that the evoked dopamine release in the NAc between the rat pups exposed to nicotine and the control pups is noticeably different. The DA output of the nicotine-exposed pup shows a greater release following optogenetic stimulation than that of the control animal, despite a delayed reaction to the stimulation from the control pup. The intensity of DA release with regards to the percent baseline level is substantially higher at each time interval, meaning that the maturation of the mesolimbic pathway is indeed altered due to the chronic presence of nicotine in the rat pups VTA pre- and post-gestation.

## 4. Conclusions

In conclusion, our optogenetics and microdialysis experimental system has a unique advantage in that it can be applied to smaller test subjects, including juvenile rats, to investigate the maturation of neural networks. In this study, we tested the efficacy of our system by obtaining DA concentration output from pups. To do so, we stimulated the VTA using a PS device in conjunction with the ChrimsonR AAV and incorporated microdialysis with HPLC-ECD for sample analysis. Based on our obtained data, we concluded that we successfully stimulated the VTA by comparing it to previous work and identifying the characteristic trend for DA release following excitation [14,15,16,17,18]. Our preliminary results furthermore showed, at least for P14, that the DA increased release in the NAc is due to nicotine exposure. Our experiment is challenged by a small sample size and the combined excitation of both GABAergic and dopaminergic neurons. Additionally, it does not aim to quantify the toxicity of nicotine after intrauterine exposure. Therefore, we plan to implement genetically modified DAT-Cre rats injected with the ChrimsonR AAV and a higher sample size in future experiments.

## 5. Materials and Methods

### 5.1. Ethics Statement and Perinatal Nicotine Administration

All experimental protocols and surgical procedures approved by the University of Houston Animal Care Operations (ACO) and the Institutional Animal Care and Use Committee (IACUC) were performed per accepted guidelines and regulations and complied with the ARRIVE guidelines.

In this study, 6 pregnant female wild-type Sprague-Dawley (SD) rats were purchased from Charles River (Charles River, Wilmington, MA, USA), and we obtained them on gestational day 3. They were allowed 72 h to acclimate in the animal facility, where they were maintained at 22 ± 2 °C with 65% humidity on a 12-h light/12-h dark cycle before any procedures were performed. On gestation day 6, 4 pregnant rats deemed to be nicotine-exposed mothers received treatment via an osmotic pump (Alzet, Cupertino, CA) containing nicotine hydrogen tartrate (Sigma-Aldrich, St. Louis, MO, USA) released at a rate of 6 mg/kg/day to simulate a moderate to heavy smoker. Additionally, 2 pregnant rats were deemed to be control moms which were not exposed to nicotine. The pump was surgically implanted using a posterior subcutaneous incision where it would diffuse the respective substance for 28 days, from gestational day 6 to postnatal day 14. The pups were kept with their mothers from postnatal day 1 to day 21, and, thus, they were provided with food and water ad libitum. Pups were monitored for 7 days before surgery was performed. On postnatal day 21, we separated pups by sex, and the males were placed in a separate cage.

### 5.2. PS Device and Fabrication

The PS device, as used in previous experiments, is fabricated with a flexible printed circuit (FPC) made with polyimide as the foundation for the device [15]. Using epoxy (Z-1, NISSIN RESIN Co., Ltd., Japan), the LEDs (ES-AEHRAX12, Epistar Corporation, Taiwan) are attached to the FPC in a column, and each LED is 300 µm wide and 300 µm long. The LED light intensity is controlled via the light pulse system, and each LED has a wavelength of 620 nm, the necessary wavelength to activate ChrimsonR. The FPC was baked to set the epoxy, and the LEDs were wire bonded to the FPC. The FPC was then covered in epoxy to preserve the connections and avoid damage to brain tissue during implantation. Finally, the device was coated in biocompatible parylene-C to protect it from water damage and to provide electrical insulation.

### 5.3. Stereotaxis Surgery

Each pup received the following two surgeries: (1) AAV injection; and (2) Microdialysis cannula and PS device implantation. The surgical procedures for virus injection, microdialysis cannula implantation, and PS device in rats as detailed in our previous paper [15].

### 5.4. AAV Injection

On P7, pups underwent AAV injection surgery. We injected pAAV-CamKIIa-ChrimsonR-mScarlet-KV2.1 (Addgene, Watertown, MA, USA) into the VTA (location calculated from literature [24]) to achieve virus-mediated expression of ChrimsonR, which is a light-gated cation channel protein capable of activating dopaminergic neurons. The virus was injected at 0.1 µL/min for 10 min, and the virus required 7 days for full expression in the VTA.

P7 pups were anesthetized using isoflurane, and their head was fixed in a stereotaxic surgery station for rats (Narishige, Tokyo, Japan). An incision was made along the sagittal suture to expose the skull. A bur hole was made above the VTA, and the dura was broken using a glass capillary needle. The virus injection needle was then slowly inserted to the correct depth, and the injection began. Following the completion of the injection, the injection needle remained for an additional 5 min in the VTA to ensure complete administration. The needle was then slowly removed, and a suture (Ethicon, Raritan, NJ, USA) was used to close the incision. The pup was removed from isoflurane, administered buprenorphine (Henry Schein, Melville, NY, USA), and monitored until fully awake. We then returned the pup to its mother.

### 5.5. Microdialysis Cannula and PS Device Implantation

Many widely used optogenetic tools are too damaging, lack the ability for implantation, or hinder freely moving experiments. Despite these limitations, much research has been conducted, including work with optical fibers. Optical fibers allow accurate stimulation in a localized area within the deep brain with minor tissue damage [25]. Despite the strengths of optical fibers, the main weakness lies in extreme rigidity, which potentially hinders freely moving experiments. Additionally, chronic studies propose damage risks to optical fibers due to their fragility. Sample collection is also crucial in measuring and interpreting DA output, but the collection method must comply with the delicate nature of the pup’s brain and skull [25]. Therefore, alternative means for optical stimulation and sample collection are needed. Our PS device is flexible, thin, and minimizes damage caused to brain tissue upon implantation. Its power source is delivered via a thin cable and is lightweight, easily facilitating freely moving experimental conditions, and the microdialysis system features similar characteristics [12,14,15].

We implanted the microdialysis cannula and the PS device into the P13 pup. The animal was sedated using a combination of ketamine (Covetrus, Portland, ME, USA) and xylazine (Covetrus) and was transferred to the stereotaxic surgery station (Narishige). The skull was exposed, and bur holes were made above the NAc and VTA. After the dura was broken, the PS device was slowly inserted into the VTA, and the microdialysis cannula was then positioned above the NAc and slowly inserted [24]. A dummy probe was inserted (EICOM, San Diego, CA, USA). The implantation was then fixed to the skull using dental composite resin [Sun Medical Co., Shiga, Japan] to finalize the procedure. The pups were provided with buprenorphine [Henry Schein], monitored until they awoke, and then returned to their home cage with their mother.

### 5.6. Investigating Efficacy of Experimental System Using Juvenile Rat Pups

Dopaminergic and GABAergic neurons in the VTA were activated by exciting ChrimsonR in wild-type rat pups (*n* = 10) using light (red, 620 nm) produced by a PS device. We implanted the PS device so the LEDs faced medially, exposing light to the VTA. Simultaneously, we collected dialysate from the NAc through an implanted microdialysis cannula, where we inserted the microdialysis probe. This dialysate was then measured using HPLC-ECD, producing precise voltage graphs that identified DA concentration over time. This experiment was performed under freely moving conditions.

### 5.7. Optogenetics Stimulation Setup

Before beginning the experiment, the PS device implanted into the P14 rat was connected to its respective cable, and we set the stimulation parameters. (Figure 5) shows the pulse cycle where the stimulation lasted 5 min total, alternating a pattern of 10 s on and 10 s off. We set the pulse intensity to 3 mA with a period of 500 ms and a pulse width of 100 ms to repeat 20 times. Following the 20th repetition, there was 10 s of no stimulation. The entire cycle would then repeat 15 times. We monitored the light intensity to allow activation of ChrimsonR without causing heat damage to the brain.

### 5.8. Optogenetics and Microdialysis Experiment

To focus on maturation and our unique system, we began our experiment with nicotine-exposed pups starting from P14. They were allowed more than one night to recover after the implantation of the microdialysis cannula and the PS device before proceeding with the freely moving experiment. To initiate the experiment, the dummy probe placed in the implanted microdialysis cannula was replaced with the microdialysis probe while the animal was under isoflurane anesthesia and, using flowing Ringer’s solution, the probe’s membrane collected the dialysate containing DA from the NAc. After the PS device and microdialysis probe were connected, we placed the animal in an enclosure with freely moving conditions. The experiment then commenced with Ringer’s solution being pumped at 1 µL/min through the microdialysis probe, allowing the perfusion of the solution to pass through the NAc, with dialysates collected at 15-min intervals. For sample analysis, 10 µL dialysate was evaluated from each sample using HPLC-ECD. With this method, the concentration of DA could be measured. Preprepared DA standards of 0.1, 0.5, and 1 pg/µL were used to provide calibration for the HPLC machine.

Initial DA concentration is somewhat abnormal due to handling and residual effects from isoflurane anesthesia. There is often a steady decline until a consistent DA concentration is achieved, from which an average is taken and used as the control. After approximately 2 h, the animal’s behavior and DA levels stabilized, and we initiated the stimulation. Once stabilized, stimulation was given, which lasted for the initial 5 min of the ongoing 15-min sample, and 6–8 samples were obtained after initial stimulation, as depicted in (Figure 6).

### 5.9. Statistical Analysis

Statistical analysis was performed on comparison data between different age groups and between test groups and control groups. One-way analysis of variance (ANOVA) followed by Bonferroni’s post hoc correction and paired *t*-tests were utilized to evaluate statistical significance. For any *p*-value < 0.05, the results are deemed to differ significantly.

## Figures and Tables

**Figure 1 ijms-24-02280-f001:**
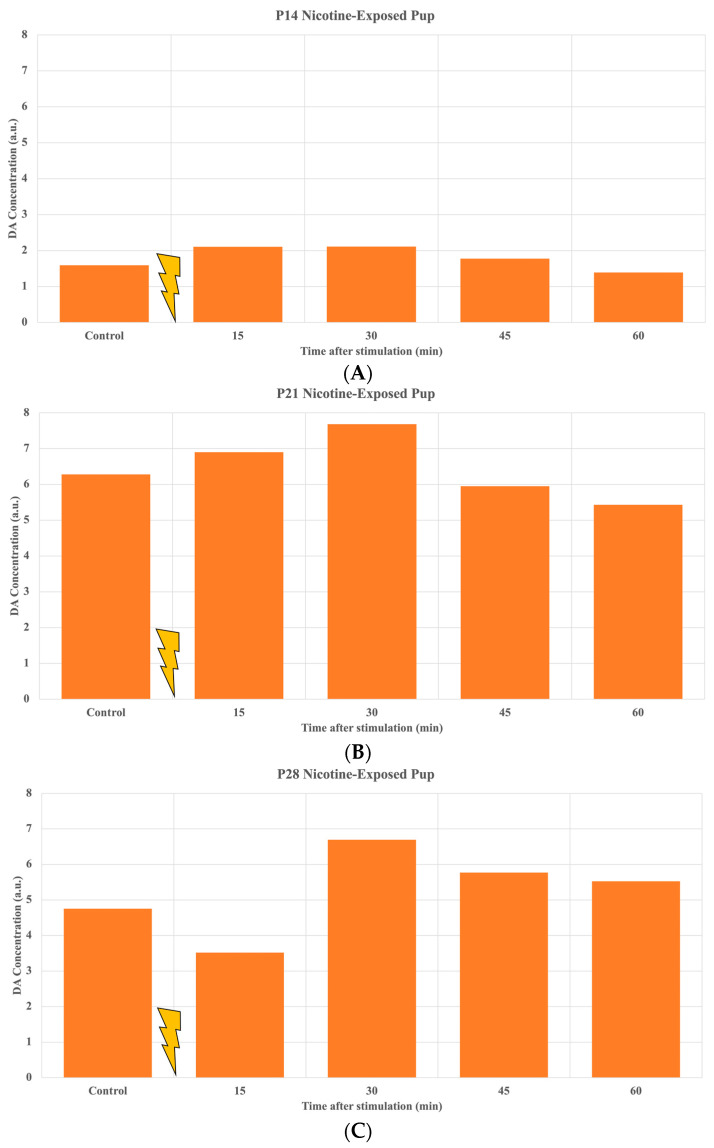
DA in NAc before (control) and after (0–60 min) stimulation of the three representative nicotine-exposed rat pups including P14 (**A**), P21 (**B**), and P28 (**C**). Our data demonstrate the increase in DA release into NAc following stimulation of VTA.

**Figure 2 ijms-24-02280-f002:**
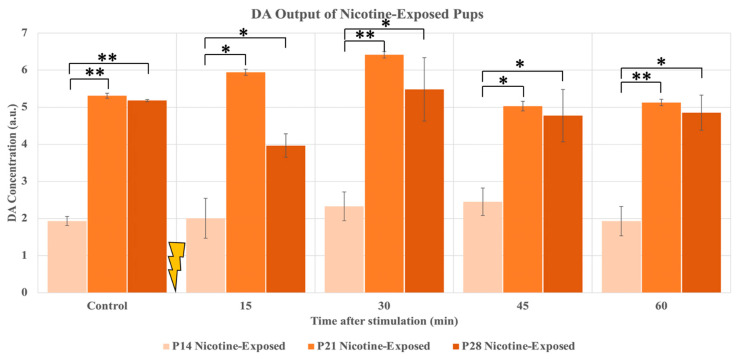
DA output of rat pups (P14, P21, P28) exposed to nicotine perinatally. Results are compared measurements of raw DA output into the extracellular space of the NAc. A significant difference was observed between the P14 pups and the P21 and P28 pups. ** denotes *p* < 0.01, ±standard deviation. * denotes *p* < 0.05, ±standard deviation.

**Figure 3 ijms-24-02280-f003:**
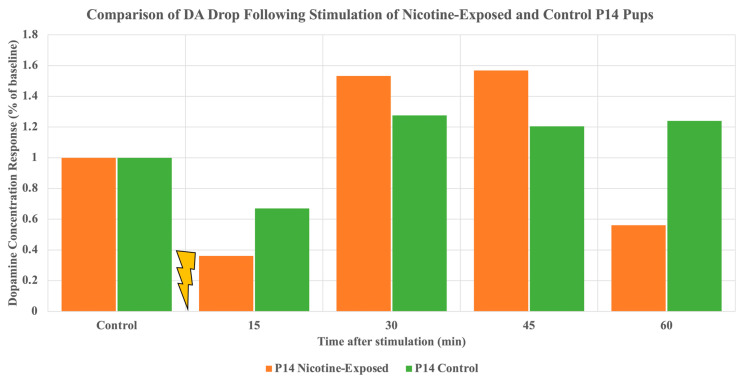
Depiction of correlation of post-stimulation drop of normalized DA output of two P14 pups: one perinatally exposed to nicotine and one having no nicotine exposure (control). The characteristic drop in DA output to the NAc is observed at the 15-min mark following immediately after stimulation. A rise in DA output is observed at the 30-min mark following the drop, leading to the hypothesis that GABA neurons may have caused the rapid suppression of dopaminergic neurons.

**Figure 4 ijms-24-02280-f004:**
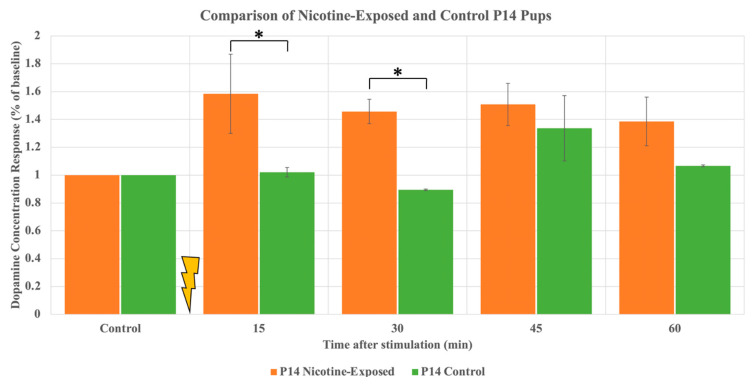
Comparison of normalized DA output of P14 pups: those perinatally exposed to nicotine and those having no nicotine exposure (control). The DA output of a nicotine-exposed pup is observed to be greater than that of the control pup, leading to the inference that the mesolimbic pathway of the nicotine-exposed pups is altered in comparison the pups not exposed to nicotine. * denotes *p* < 0.05, ±standard deviation.

**Figure 5 ijms-24-02280-f005:**
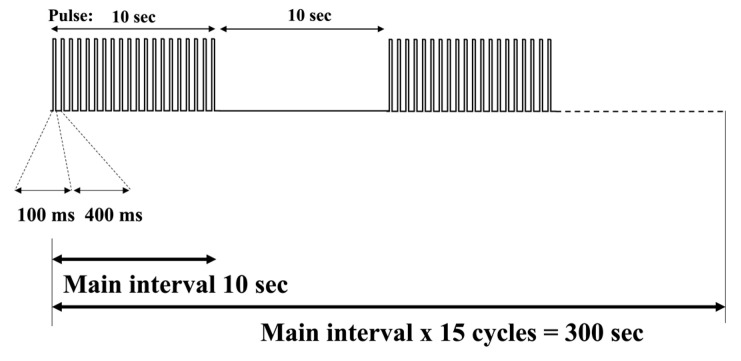
Stimulation duration consisted of 15 consecutive cycles amounting to 5 min of total stimulation. Each cycle consisted of two parts. The first half included the pulsing light, with each pulse consisting of 100 ms of light and 400 ms of no light. This is repeated 20 times for a total of 10 sec. The second half consisted of no stimulation, meaning no pulsing light. This lasted for 10 s, combining for a total of 20 s *per cycle*.

**Figure 6 ijms-24-02280-f006:**
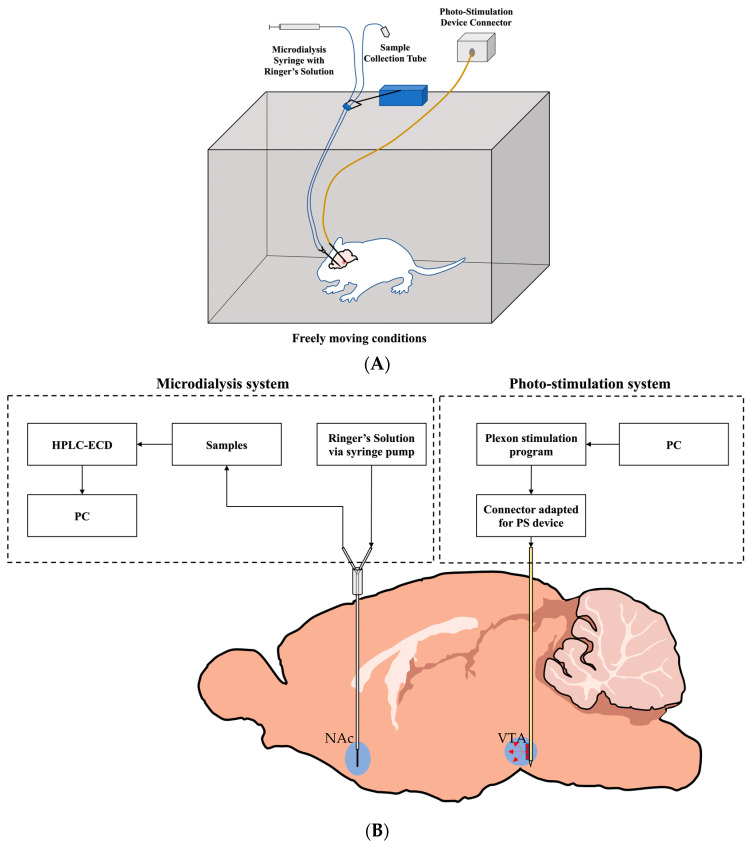
Visualization of the general experimental setup for rat pups. (**A**) Experimental setup diagram depicting a rat pup in a freely moving environment. Both the microdialysis cannula and the PS device are implanted. The microdialysis probe and tubes are collecting dialysate and the PS device is connected to a power supply via a customized connection cable acting as the device connector. (**B**) Visualization of stimulation of the mesolimbic pathway starting with stimulation in the VTA and ending with dialysate collection in the NAc.

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
