# Peer review of "Investigating the Modulation of the VTA Neurons in Nicotine-Exposed Pups during Early Maturation Using Optogenetics"

_ijms, 2023, doi:10.3390/ijms24032280_

Round 1
Reviewer 1 Report
Ganaway et al investigated the influence of stubstance abuse on newborn pups exposed to nicotine.
The manuscript is well written and the topic is very interesting. I have only a minor comment:
1)Differences showed in figure 3 and, are statistically significant? Please add symbols on the graph to highlight when p<0.05
Author Response
Reviewer 1
- Differences showed in figure 3 and, are statistically significant? Please add symbols on the graph to highlight when p<0.05
Statistical significance has now been included in the manuscript.

Reviewer 2 Report
The manuscript entitled Investigating the Modulation of the VTA Neurons in Nicotine-Addicted Pups During Early Maturation Using Optogenetics addresses an interesting topic and describes a technique that can be used in future reproductive toxicity studies. However, I have a few observations:
The authors do not specify how many pregnant rats were included in the study. For reproductive toxicity tests, according to OECD (OECD Guideline for Testing of Chemicals. Prenatal Developmental Toxicity Study), at least 20 pregnant females/group are required.
I understood from the manuscript (including the abstract) that the article describes a technique and does not aim to quantify the toxicity of nicotine after intrauterine exposure. This should also be specified in the conclusions section also, which is missing in the current form of the manuscript (please add the conclusion section).
The references are not in accordance with the style required by the journal.
The technique can be improved, as the authors also stated, but I appreciate the team's effort to develop a new technique.
Author Response
Reviewer 2:
- The authors do not specify how many pregnant rats were included in the study. For reproductive toxicity tests, according to OECD (OECD Guideline for Testing of Chemicals. Prenatal Developmental Toxicity Study), at least 20 pregnant females/group are required.
We have addressed this by adding the information about the mother rats to the animal care section in the materials and methods.
- I understood from the manuscript (including the abstract) that the article describes a technique and does not aim to quantify the toxicity of nicotine after intrauterine exposure. This should also be specified in the conclusions section also, which is missing in the current form of the manuscript (please add the conclusion section).
We have added the conclusion section.
- The references are not in accordance with the style required by the journal.
The references have been adjusted to the required style.

Reviewer 3 Report
The manuscript entitled ‘Investigating the Modulation of the VTA Neurons in Nicotine-Addicted Pups During Early Maturation Using Optogenetics’ by Ganaway et al, investigates the effects on evoked dopamine release in the NAc in rat pups perinatally exposed to nicotine. The study addresses the important topic of nicotine mediated mesolimbic circuit maturation and ultimately dopamine output in the NAc. However, there are some serious issues with the current state of the manuscript.
1) Statistical tests have not been used nor reported throughout the manuscript to explain results.
2) Controls are missing for the experiments. That is to compare evoked dopamine release in the NAc between rat pups exposed to nicotine vs nicotine naïve animals. Therefore, it is not clear how nicotine exposure alters mesolimbic circuit maturation and dopamine output. This has been partially reported in Figure 5. However, no comparison was made between the nicotine naïve pup (control) and nicotine exposed pup.
3) There was no reference to Figure 5 in Results section.
4) The authors throughout the paper call nicotine exposed pups as nicotine addicted pups when this wasn’t specifically tested and reported.
5) Sections 2.1- 2.3 (Figures 1 & 2) should have been written in Materials & methods, not Results section.
6) Technically only one experiment was conducted in this whole study. The authors could have complemented with other techniques, for example, electrophysiological recordings from neurons in the VTA and NAc in control and nicotine exposed pups. The authors should have also provided histological proof of virus expression in the VTA.
7) The current study suffers from a small sample size and non-specific excitation of VTA neurons, limitations which were identified by the authors in the Discussion section. However, these issues should have been addressed and implemented in the current study itself.
Author Response
Reviewer 3:
- Statistical tests have not been used nor reported throughout the manuscript to explain results.
Statistical methods have now been included in the manuscript.
- Controls are missing for the experiments. That is to compare evoked dopamine release in the NAc between rat pups exposed to nicotine vs nicotine naïve animals. Therefore, it is not clear how nicotine exposure alters mesolimbic circuit maturation and dopamine output. This has been partially reported in Figure 5. However, no comparison was made between the nicotine naïve pup (control) and nicotine exposed pup.
We have included further information regarding the comparison of the nicotine-exposed and the control pups. Although the purpose of this manuscript is to convey efficacy of our system, we agree that it is beneficial to provide preliminary data regarding control P14 pups as well.
- There was no reference to Figure 5 in Results section.
We have adjusted the manuscript accordingly.
- The authors throughout the paper call nicotine exposed pups as nicotine addicted pups when this wasn’t specifically tested and reported.
We have made the recommended change from “nicotine-addicted” to “nicotine-exposed”.
- Sections 2.1- 2.3 (Figures 1 & 2) should have been written in Materials & methods, not Results section.
Thank you for the correction. We have adjusted the manuscript accordingly.
- Technically only one experiment was conducted in this whole study. The authors could have complemented with other techniques, for example, electrophysiological recordings from neurons in the VTA and NAc in control and nicotine exposed pups. The authors should have also provided histological proof of virus expression in the VTA.
Additional information has been included in the introduction to emphasize previous electrophysiological experimentation from our lab and similar external studies to provide foundational context for our current experimental procedure featured in this manuscript.
In the discussion section, we compared our findings regarding successful virus expression and VTA stimulation to previous results achieved from both our lab and external studies to identify the characteristic DA trend following stimulation of the VTA to confirm that the virus expression in our experiment was successful.
- The current study suffers from a small sample size and non-specific excitation of VTA neurons, limitations which were identified by the authors in the Discussion section. However, these issues should have been addressed and implemented in the current study itself.
Thank you for the highlighting these points. The overarching purpose of this manuscript is to present proof of efficacy regarding our novel stimulation and measurement platform for the analysis of maturing rat pups. The experimental procedure is very difficult due to the fragility of the animal and its size. Additionally, it takes at least 2-3 months to complete one cycle starting with the pregnant mother and ending with a nearly completely mature rat. Therefore, we set out to establish efficacy of our experimental system with the intent to expand on the findings in this manuscript with several new experiments in the near and distant future. Ultimately, we believe this study will stimulate the research of other groups and encourage said groups to consider using optical stimulus for not only the investigation of the mesocorticolimbic system, but also for any form of neurostimulation research.

Round 2
Reviewer 3 Report
The revised manuscript has some improvements from its previous version. However, the authors should address some of the following points to strengthen the manuscript.
1) For statistics, please specify which post-hoc test was used with 1 way ANOVA. Also please report the t or F statistic in Results section along with exact p value.
2) In section 2.1, please use statistics to compare DA output measured at different time points to control across the ages. Especially since authors claim that a ‘significant increase in DA release in the NAc’ was observed.
3) In Figure 1 captions, number of rat pups used for the study is reported as 15. This is a huge change from the number reported in the previous manuscript version (n=3). Were more pups added in the study? Please clarify.
4) Fig 2 Y-axis says % of baseline. Please correct. What is the number of animals used in this study? Please specify in captions.
5) Can the authors speculate that the changes in DA output across p14, p21 and p28 be due to changes in virus expression rather than physiological changes?
6) Was data from the one nicotine exposed pup in Fig 3 included in average data shown in Fig 1? Can the authors clarify why there are error bars in bar graphs in Fig 3 if data is derived from only one animal in both groups?
7) In Fig 4, why is the trend of evoked DA output in p14 nicotine exposed pup so different from the ones shown in Fig 1 & 2? Why is there no increase in evoked DA release in the control pups even though such a trend was shown in Fig 3?
8) Please specify number of animals used for Fig 4 results in captions. Why weren’t the groups tested at p21 and p28?

Author Response
Reviewer 3
- For statistics, please specify which post-hoc test was used with 1 way ANOVA. Also please report the t or F statistic in Results section along with exact p value.
The post-hoc test has now been included in the manuscript along with t-test.
Please see lines 396-400. Please see the results section.
- In section 2.1, please use statistics to compare DA output measured at different time points to control across the ages. Especially since authors claim that a ‘significant increase in DA release in the NAc’ was observed.
We used only 1 animal and 1 stimulation for each subfigure in Figure 1 to demonstrate effective stimulation of the VTA using optogenetics. Consequently, statistical measures are no longer needed.
- In Figure 1 captions, number of rat pups used for the study is reported as 15. This is a huge change from the number reported in the previous manuscript version (n=3). Were more pups added in the study? Please clarify.
We revised the figures in the paper. Figure 1. shows DA in NAc before (control) and after (0-60 min) stimulation of three representative nicotine-exposed rat pups including P14 (A), P21 (B), and P28 (C). Our data demonstrates the increase in DA release into NAc following stimulation of VTA.
Therefore, we only used a total of 3 pups (with single stimulus) in this figure.
- Fig 2 Y-axis says % of baseline. Please correct. What is the number of animals used in this study? Please specify in captions.
We have corrected the Y-axis unit for Figure 2.
We used a total of 6 rat pups (3 for p14, 2 for P21, and 1 for P28). Furthermore, we stimulated each animal at least twice.
- Can the authors speculate that the changes in DA output across p14, p21 and p28 be due to changes in virus expression rather than physiological changes?
While virus expression may play a role in the changes in DA output, the changes would be considered negligible. Using previous studies of both adult and adolescent rats and mice produced by our lab and others, we can infer this. In our future studies, we plan to introduce combination DAT-Cre and ChrimsonR transgenic animals into our study. In this case, virus injection would not be necessary. Therefore, it would eliminate potential consequences of virus expression. Thank you for bringing up this important point. We have adjusted the manuscript to include this information.
Please see lines 219-221 for details.
- Was data from the one nicotine exposed pup in Fig 3 included in average data shown in Fig 1? Can the authors clarify why there are error bars in bar graphs in Fig 3 if data is derived from only one animal in both groups?
We agree with the reviewer. We used two new representative rat pups (P14) including one for nicotine-exposed and one for control.
- In Fig 4, why is the trend of evoked DA output in p14 nicotine exposed pup so different from the ones shown in Fig 1 & 2? Why is there no increase in evoked DA release in the control pups even though such a trend was shown in Fig 3?
For figure 4, we only observed the DA increase after 45 minutes for the controls.
- Please specify number of animals used for Fig 4 results in captions. Why weren’t the groups tested at p21 and p28?
As we indicated before, the overarching purpose of this manuscript is to present proof of efficacy regarding our stimulation and measurement platform for the analysis of maturing rat pups. Here we just want to show, at least for P14, that the increased DA in the NAc is due to the nicotine exposure. We plan to expand in our future studies with increase sample size.
